# Digital Transformation in Healthcare: Technology Acceptance and Its Applications

**DOI:** 10.3390/ijerph20043407

**Published:** 2023-02-15

**Authors:** Angelos I. Stoumpos, Fotis Kitsios, Michael A. Talias

**Affiliations:** 1Healthcare Management Postgraduate Program, Open University Cyprus, P.O. Box 12794, Nicosia 2252, Cyprus; 2Department of Applied Informatics, University of Macedonia, 156 Egnatia Street, GR54636 Thessaloniki, Greece

**Keywords:** digital transformation, bibliometric, digitalisation, e-health, m-health, telemedicine, personalised healthcare, artificial intelligence, acceptance of technology, security

## Abstract

Technological innovation has become an integral aspect of our daily life, such as wearable and information technology, virtual reality and the Internet of Things which have contributed to transforming healthcare business and operations. Patients will now have a broader range and more mindful healthcare choices and experience a new era of healthcare with a patient-centric culture. Digital transformation determines personal and institutional health care. This paper aims to analyse the changes taking place in the field of healthcare due to digital transformation. For this purpose, a systematic bibliographic review is performed, utilising Scopus, Science Direct and PubMed databases from 2008 to 2021. Our methodology is based on the approach by Wester and Watson, which classify the related articles based on a concept-centric method and an ad hoc classification system which identify the categories used to describe areas of literature. The search was made during August 2022 and identified 5847 papers, of which 321 fulfilled the inclusion criteria for further process. Finally, by removing and adding additional studies, we ended with 287 articles grouped into five themes: information technology in health, the educational impact of e-health, the acceptance of e-health, telemedicine and security issues.

## 1. Introduction

Digital transformation refers to the digital technology changes used to benefit society and the healthcare industry. Healthcare systems need to use digital technology for innovative solutions to improve healthcare delivery and to achieve improvement in medical problems. The digital transformation of healthcare includes changes related to the internet, digital technologies, and their relation to new therapies and best practices for better health management procedures. The quality control of massive data collected can help improve patients’ well-being and reduce the cost of services. Digital technologies will also impact medical education, and experts will deceive new ways to train people. Now in this way, practitioners will face new opportunities. 

Digital transformation is an ongoing process that can create opportunities in the health sector, provided the necessary infrastructure and training are available. Under Regulation (EU) 2021/694 of the European Parliament and of the Council of 29 April 2021, establishing the Digital Europe Program and repealing Decision (EU) 2015/2240, digital transformation is defined as the use of digital technologies for the transformation of businesses and services. Some technologies that contribute to digital transformation are the digital platform of the Internet of Things, cloud computing and artificial intelligence. At the same time, the sectors of society that are almost affected are telecommunications, financial services and healthcare.

Digital health can play a role in innovation in health, as it facilitates the participation of patients in the process of providing health care [1]. The patient can overcome his poor state of health when they are no longer in a state of well-being. In this case, the patient is given the to participate in the decision-making regarding their health care. Searching for information through the patient’s internet or using digital health applications (e.g., via mobile phone) is essential for the patient to make the right decision about their health.

In the coming years, health change is expected to focus primarily on the patient, who will take on the “health service consumer” role as the patient seeks control over their health management. The healthcare industry will be shaped based on the needs and expectations of this new “consumer of health services”, which will require upgraded experiences with the main characteristics of personalisation, comfort, speed and immediacy in the provision of services. Gjellebaek C. et al. argue that new digital technologies will shift healthcare towards digitalisation, bringing significant benefits to patients and healthcare infrastructure [2]. Some of the benefits listed by Gjellebaek C. are the increase in employee productivity, the improvement of the efficiency and effectiveness of the operation of the health units, and the reduction of their operating costs.

On the other hand, in terms of health infrastructure, a typical example is the United States, where 75% of hospitals use electronic health record systems, according to Rebekah E. et al. [3]. However, clinicians often report side effects using digital technologies, which can be attributed to their misuse [3]. In addition, some health professionals oppose using these systems and develop solutions that jeopardise patient care. In some countries, such as the United States, the government provides incentives for the “effective use” of e-health technologies, but their results remain uncertain [3].

Rebekah E. et al. focus more specifically on U.S. hospitals, observing that the remaining countries are relatively in the early stages of transformation [4]. The United Kingdom, for example, has recently pursued troubled e-health initiatives, and Australian hospitals have only recently participated in investments in the digitalisation of their hospital services [4]. At the European Union level, digital health is a critical key strategic priority, in line with the European Strategic Plan 2019–2024 (European Commission).

Today, digital transformation in health is spreading and consolidating rapidly [5]. The purpose of this paper is to provide an assessment of the current literature on digital health transformation, as well as to identify potential vulnerabilities that make its implementation impossible. The ultimate goal is to see how digital technologies facilitate patients’ participation in health and their health.

Due to the rapid development of e-health and digitalisation, data from previous studies are becoming potentially irrelevant. Most studies evaluating digitalisation have relied heavily on quantitative research-based methods. Although quantitative evaluations are required, some of their effects could be omitted.

According to Gopal G. et al., healthcare has the lowest level of digital innovation compared to other industries, such as media, finance, insurance and retail, contributing to limited labour productivity growth [6]. With this article, we seek to reverse this picture and contribute to the emergence of digitalisation as a factor of health innovation while optimising patient outcomes and the cost of services provided. However, to achieve this innovation, systemic changes are needed in healthcare finances, the education of healthcare staff and healthcare infrastructure.

The following section analyses the methodology and its steps, which then contributed to the emergence of our results.

## 2. Material and Methods

### 2.1. Search Strategy and Bibliography Reviews

Our research approach is based on the methodology of Webster and Watson, who developed a concept-centric method and an ad hoc classification system in which categories are used to describe areas of literature [7]. Initially, the existing bibliographic reviews were searched to select the databases based on keywords. A retrospective search was then performed to examine the reports of the selected works. Finally, the references of selected works were investigated to increase the search sample through the future search. After selecting the articles, they were grouped according to their content.

Systematic reviews were conducted to place this paper on existing knowledge of digital health, as well as to review prior knowledge in this area and to discuss recognised research questions based on the results of previous studies. A comprehensive review of the published literature was reported by Marques, I. C., & Ferreira, J. J. [8]. The authors explored the potential of existing digital solutions to improve healthcare quality and analysed the emerging trend in digital medicine to evaluate the research question of how stakeholders apply and manage digital technologies for business purposes [9]. The main question is: How and what could be done sustainably and inclusively through innovation to achieve sustainable development goals by taking advantage of Information and Communication Technologies? Recently, researchers have expressed concern about secure communication and user authentication within providing information to patients. In contrast with data storage, information exchange, and system integration, new approaches and uses of patient care processes are envisaged with the prospect of monitoring not only diagnostic statistics but also in-depth analysis of signs and symptoms before and after treatment, essential sources for new research. Table 1 presents the previous bibliographic reviews on which our study was based. 

### 2.2. Network Analysis

Network analysis is considered a branch of graph theory. Our network analysis is based on the similarity of keywords found in identifying the eligible papers. We used visualisation of similarities (VOS) software, version 1.6.18, to construct graphical networks to understand the clustering of the keywords and their degree of dissimilarity. Our network analysis is based on the similarity of keywords found in identifying the eligible papers.

#### Initial Search

The search was performed on the following databases: Scopus, Science Direct, and PubMed, using the keywords “digital transformation”, “digitalisation”, “Ehealth or e-health”, “mhealth or m-health”, “healthcare” and “health economics”. We selected publications from the search of international journals and conference proceedings. We collected papers from 2008 until 2021. The documents sought belonged to strategy, management, computer science, medicine, and health professions. Finally, the published works were in English only. The total number of articles collected using the keywords as shown in Table 2 was 5847. 

We systematically checked the total number of papers 5847 by reading their titles, abstracts, and, whenever necessary, the article’s first page to conclude if each document was relevant as a first step as shown in the Figure 1.

Then, we looked at the titles of the 378 articles, and after reading their summary, we accepted 321 articles. Further studies were rejected because their full text was not accessible. As a result, there were 255 articles in our last search. Of the selected 255 articles, 32 more were added based on backward and forward research. The investigation was completed by collecting common standards from all databases using different keyword combinations. According to the systematic literature review, we follow the standards of Webster and Watson (2002) to reject an article. Since then, we have collected the critical mass of the relevant publications, as shown in Figure 2.

## 3. Results

### 3.1. Chronological Development of the Publications

The categorisation of the articles was based on their content and the concepts discussed within them. As a result, we classify articles into the following categories: information technology in health, the educational impact on e-health, the acceptance of e-health, telemedicine, and e-health security. 

Although researchers in Information and Communication Technology and digitalisation conducted studies almost two decades ago, most publications have been published in the last eight years. This exciting finding highlights the importance of this field and its continuous development. Figure 3 shows a clear upward trend in recent years. More specifically, the research field of Information and Communication Technology, in combination with digital transformation, appeared in 2008. However, the most significant number of articles was found in 2019, 2020 and 2021. The number of articles decreased to the lowest in 2009–2011 and 2013–2014. Due to the expansion of the field to new technologies, the researchers studied whether the existing technological solutions are sufficient for implementing digital transformation and what problems they may face.

Figure 3 shows a combination of the articles per year and the number of citations per publication per year.

### 3.2. Document Type

Of the document types, 59.51 per cent of the articles were categorised as “survey”, while a smaller percentage were in: “case study” (32.53%), “literature review” (5.88%) and “report” (2.08%). However, these documents focused on specific concepts: “information technology in health” (45%), “education impact of e-health” (11%), “acceptance of e-health” (19%), “telemedicine” (7%), “security of e-health” (18%).

As we can see from the following Figure 4, we used network analysis, where the keywords related to digitalisation and digital transformation were identified in the research study. Network analysis, using keywords, came with VOSviewer software to find more breadth and information on healthcare digitalisation and transformation exploration. It was created by analysing the coexistence of keywords author and index. This analysis’s importance lies in the structure of the specific research field is highlighted. In addition, it helped map the intellectual structure of scientific literature. Keywords were obtained from the title and summary of a document. However, there was a limit to the number of individual words. The figure represents a grid focused on reproducing keywords in the literature on the general dimensions of digitalisation. The digitalisation network analysis showed that e-health, telemedicine, telehealth, mobile health, electronic health/medical record, and information systems were the main relevant backgrounds in the literature we perceived. In the healthcare literature, keywords such as “empowerment” and “multicenter study” usually do not lead to a bibliographic search on digitalisation. Figure 4 shows how e-health and telemedicine have gone beyond the essential and most crucial research framework on how they can affect hospitals and the health sector. The potentially small gaps in network analysis can be filled by utilising data in our research study, contributing to future research.

Figure 5 shows the network analysis with the keywords concerning time publication. The yellow colour indicates keywords for most recent years.

Figure 6 presents the density visualisation of keywords.

Figure 7 shows the number of articles per each method (survey, literature review etc.) for each year. 

It is evident from Figure 7 that the most used method paper is the survey type and that in the year 2021, we have a high number of surveys compared to previous years. 

### 3.3. Summary of the Included Articles

In Figure 2, we have explained how we collected the critical mass of the 255 relevant publications. We added another 32 articles based on further research with the backward and research methods, which resulted in a total number of 287 articles.

Then, the articles were categorised according to their content. The concepts discussed in the papers are related to information technology in health, the educational impact of e-health, the acceptance of e-health, telemedicine, and e-health security. For this purpose, the following table was created, called the concept matrix table. 

## 4. Concept Matrix

In this section, we provide the Concept matrix table. Academic resources are classified according to if each article belongs or not to any of the five concepts shown in Table 3.

## 5. Analysis of Concepts

From the articles included in the present study between 2008 and 2021, they were grouped into five categories identified: (i) information technology in health, (ii) acceptance of e-health, (iii) telemedicine, (iv) security of e-health, and (v) education impact of e-health.

### 5.1. Information Technology in Health

Researchers have studied several factors to maximise the effectiveness and success of adopting new technology to benefit patients. Hospitals can benefit from information technology when designing or modifying new service procedures. Health units can use information and communication technology applications to analyse and identify patients’ needs and preferences, enhancing their service innovation processes. Previous findings conclude that technological capability positively influences patient service and innovation in the service process [301]. These results have significant management implications as managers seek to increase technology resources’ efficiency to achieve patient-centred care as the cornerstone of medical practice [207].

Informatics facilitates the exchange of knowledge necessary for creating ideas and the development process. The internet supports health organisations in developing and distributing their services more efficiently [206]. Also, Information Technology improves the quality of services, reduces costs, and helps increase patient satisfaction. As new technologies have created opportunities for companies developing high-tech services, healthcare units can increase customer value, personalise services and adapt to their patient’s needs [209]. To this end, the “smart hospitals” should represent the latest investment frontiers impacting healthcare. Their technological characteristics are so advanced that the public authorities need know-how for their conception, construction, and operation [228].

A new example is reshaping global healthcare services in their infancy, emphasising the transition from sporadic acute healthcare to continuous and comprehensive healthcare. This approach is further refined by “anytime and everywhere access to safe eHealth services.” Recent developments in eHealth, digital transformation and remote data interchange, mobile communication, and medical technology are driving this new paradigm. Follow-up and timely intervention, comprehensive care, self-care, and social support are four added features in providing health care anywhere and anytime [289]. However, the healthcare sector’s already precarious security and privacy conditions are expected to be exacerbated in this new example due to the much greater monitoring, collection, storage, exchange, and retrieval of patient information and the cooperation required between different users, institutions, and systems.

The use of mobile telephony technologies to support health goals contributes to the transformation of healthcare benefits worldwide. The same goes for small and medium-sized healthcare companies, such as pharmacies. A potent combination of factors between companies and customers is the reason for creating new relationships. In particular, mobile technology applications represent new opportunities for integrating mobile health into existing services, facilitating the continued growth of quality service management. Service-based, service-focused strategies have changed distribution patterns and the relationship between resellers and consumers in the healthcare industry, resulting in mobile health and significant pharmacy opportunities. It has been an important research topic in the last decade because it has influenced and changed traditional communication between professionals and patients [211]. An example of a mobile healthcare platform is “Thymun”, designed and developed by Salamah et al. aiming to create intelligent health communities to improve the health and well-being of autoimmune people in Indonesia [225].

### 5.2. Acceptance of E-Health

In a long-term project and a population study (1999–2002), Hsu et al. evaluated e-health usage patterns [302]. The authors conclude that access to and use of e-health services are rapidly increasing. These services are more significant in people with more medical needs. Fang (2015) shows that scientific techniques can be an essential tool for revealing patterns in medical research that could not be apparent with traditional methods of reviewing the medical literature [303]. Teleradiology and telediagnosis, electronic health records, and Computer-Aided Diagnosis (CAD) are examples of digital medical technology. France is an example of a country that invests and leads in electronic health records, based on what is written by Manard S. et al. [243]. However, the impact of technological innovation is reflected in the availability of equipment and new technical services in different or specialised healthcare sectors. 

On the other hand, Mariusz Duplaga (2013) argues that the expansion of e-health solutions is related to the growing demand for flexible, integrated and cost-effective models of chronic care [304]. The scope of applications that can support patients with chronic diseases is broad. In addition to accessing educational resources, patients with chronic diseases can use various electronic diaries and systems for long-term disease monitoring. Depending on the disease and the symptoms, the devices used to assess the patient’s condition vary. However, the need to report symptoms and measurements remains the same. According to Duplaga, the success of treatments depends on the patient’s involvement in monitoring and managing the disease. The emphasis on the role of the patient is parallel to the general tendency of people and patients to participate in decisions made about their health. Involving patients in monitoring their symptoms leads to improved awareness and ability to manage diseases. Duplaga argues that the widespread use of e-health systems depends on several factors, including the acceptance and ability to use information technology tools, combined with an understanding of disease and treatment.

Sumedha Chauhan & Mahadeo Jaiswal (2017) are on the same wavelength. They claim that e-health applications provide tools, processes and communication systems to support e-health practices [305]. These applications enable the transmission and management of information related to health care and thus contribute to improving patient’s health and physicians’ performance. The human element plays a critical role in the use of e-health, according to the authors. In addition, researchers have studied the acceptance of e-health applications among patients and the general public, as they use services such as home care and search for information online. The meta-analysis they use combines and analyzes quantitative findings of multiple empirical studies providing essential knowledge. However, the reason for their research was the study of Holden and Karsh (2010) [306].

To provide a comprehensive view of the literature acceptance of e-health applications, Holden and Karsh reviewed 16 studies based on healthcare technology acceptance models [306]. Findings show them that the use and acceptance of technological medical solutions bring improvements but can be adopted by those involved in the medical field.

### 5.3. Telemedicine

On the other hand, telemedicine is considered one of the most important innovations in health services, not only from a technological but also from a cultural and social point of view. It benefits the accessibility of healthcare services and organisational efficiency [215]. Its role is to meet the challenges posed by the socio-economic change in the 21st century (higher demands for health care, ageing population, increased mobility of citizens, need to manage large volumes of information, global competitiveness, and improved health care provision) in an environment with limited budgets and costs. Nevertheless, there are significant obstacles to its standardisation and complete consolidation and expansion [300].

At present, there are Telemedicine centres that mediate between the patient and the hospital or doctor. However, many factors make this communication impossible [300]. Such factors include equipment costs, connectivity problems, the patient’s trust or belief in the system or centre that applies telemedicine, and resistance to new and modern diagnostics, especially in rural and island areas. Therefore, telemedicine would make it easier to provide healthcare systems in remote areas than having a specialist in all the country’s remote regions [300]. Analysing the concept further, one can easily argue that the pros outweigh the disadvantages. Therefore, telemedicine must be adopted in a concerted effort to resolve all the obstacles we are currently facing. Telemedicine centres and services such as teleradiology, teledermatology, teleneurology, and telemonitoring will soon be included. This means that a few years from now, the patient will not have to go to a central hospital and can benefit remotely from the increased quality of health services. This will save valuable time, make good use of available resources, save patient costs, and adequately develop existing and new infrastructure. 

In 2007, the World Health Organisation adopted the following broad description of telemedicine: “The delivery of health care services, where distance is a critical factor, by all health care professionals using information and communication technologies for the exchange of valid information for the diagnosis, treatment and prevention of disease and injuries, research and evaluation, and for the continuing education of health care providers, all in the interests of advancing the health of individuals and their communities ” [307]

According to the Wayback Machine, Canadian Telehealth Forum, other terms similar to telemedicine are telehealth and e-health, which are used as broader concepts of remote medical therapy. It is appropriate to clarify that telemedicine refers to providing clinical services. In contrast, telehealth refers to clinical and non-clinical services, including education, management and research in medical science. On the other hand, the term eHealth, most commonly used in the Americas and Europe, consists of telehealth and other elements of medicine that use information technology, according to the American Telemedicine Association [308]. 

The American Telemedicine Association divides telemedicine into three categories: storage-promotion, remote monitoring, and interactive services. The first category includes medical data, such as medical photographs, cardiograms, etc., which are transferred through new technologies to the specialist doctor to assess the patient’s condition and suggest the appropriate medication. Remote monitoring allows remote observation of the patient. This method is used mainly for chronic diseases like heart disease, asthma, diabetes, etc. Its interactive services enable direct communication between the patient and the treating doctor [309]. 

Telemedicine is a valuable and efficient tool for people living or working in remote areas. Its usefulness lies in the health access it provides to patients. In addition, it can be used as an educational tool for learning students and medical staff [310]. 

Telemedicine is an open and constantly evolving science, as it incorporates new technological developments and responds to and adapts to the necessary health changes within societies.

According to J.J. Moffatt, the most common obstacles to the spread of telemedicine are found in the high cost of equipment, the required technical training of staff and the estimated time of a meeting with the doctor, which can often be longer than the use of a standard doctor [311]. On the other hand, the World Health Organisation states that telemedicine offers excellent potential for reducing the variability of diagnoses and improving clinical management and the provision of health care services worldwide. The World Health Organisation claims, according to Craig et al. and Heinzelmann PJ, that telemedicine improves access, quality, efficiency and cost-effectiveness [312,313]. In particular, telemedicine can help traditionally under-served communities by overcoming barriers to the distance between healthcare providers and patients [314]. In addition, Jennett PA et al. highlight significant socio-economic benefits for patients, families, health professionals and the health system, including improved patient-provider communication and educational opportunities [315]. 

On the other hand, Wootton R. argues that telemedicine applications have achieved different levels of success. In both industrial and developing countries, telemedicine has yet to be used consistently in the healthcare system, and few pilot projects have been able to be maintained after the end of their initial funding [316]. 

However, many challenges are regularly mentioned and responsible for the need for more longevity in many efforts to adopt telemedicine. One such challenge is the complexity of human and cultural factors. Some patients and healthcare workers resist adopting healthcare models that differ from traditional approaches or home practices. In contrast, others need to have the appropriate educational background in Information and Communication Technologies to make effective use of telemedicine approaches [314]. The need for studies documenting telemedicine applications’ economic benefits and cost-effectiveness is also a challenge. Strong business acumen to persuade policymakers to embrace and invest in telemedicine has contributed to a need for more infrastructure and program funding [312]. Legal issues are also significant obstacles to the adoption of telemedicine. These include the need for an international legal framework that allows health professionals to provide services in different jurisdictions and countries. Furthermore, the lack of policies governing data confidentiality, authentication and the risk of medical liability for health professionals providing telemedicine services [314]. In any case, the technological challenges are related to legal issues. In addition, the systems used are complex, and there is a possibility of malfunction, which could cause software or hardware failure. The result is an increase in patient morbidity or mortality as well as the liability of healthcare providers [317].

According to Stanberry B., to overcome these challenges, telemedicine must be regulated by definitive and comprehensive guidelines, which are ideally and widely applied worldwide [318]. At the same time, legislation must be enacted governing health confidentiality, data access, and providers’ responsibility [314].

### 5.4. Security of eHealth

The possibility of the patients looking at the electronic patient folder in a cloud environment, through mobile devices anytime and anywhere, is significant. On the one hand, the advantages of cloud computing are essential, and on the other hand, a security mechanism is critical to ensure the confidentiality of this environment. Five methods are used to protect data in such environments: (1) users must encrypt the information before storing it; (2) users must transmit information through secure channels; (3) the user ID must be verified before accessing data; (4) the information is divided into small portions for handling and storage, retrieved when necessary; (5) digital signatures are added to verify that a suitable person has created the file to which a user has access. On the other hand, users of these environments will implement self-encryption to protect data and reduce over-reliance on providers [210].

At the same time, Maliha S. et al. [227] proposed the blockchain to preserve sensitive medical information. This technology ensures data integrity by maintaining a trace of control over each transaction. At the same time, zero trusts provide that medical data is encrypted and that only certified users and devices interact with the network. In this way, this model solves many vulnerabilities related to data security [227]. Another alternative approach is the KONFIDO project, which aims at the safe cross-border exchange of health data. A European H2020 project aims to address security issues through a holistic example at the system level. The project combines various cutting-edge technologies in its toolbox (such as blockchain, photonic Physical Unclonable Functions, homomorphic encryption, and trusted execution) [234]. Finally, Coppolino L. et al. [271] proposed using a SIEM framework for an e-healthcare portal developed under the Italian National eHealth Net Program. This framework allows real-time monitoring of access to the portal to identify potential threats and anomalies that could cause significant security issues [271]. 

### 5.5. Education Impact of E-Health

But all this would only be feasible with the necessary education of both users and patients [11]. As the volume and quality of evidence in medical education continue to expand, the need for evidence synthesis will increase [295]. On the other hand, Brockers C. et al. argued that digitalisation changes jobs and significantly impacts medical work. The quality of medical data provided for support depends on telemedicine’s medical specialisation and knowledge. Adjustments to primary and further education are inevitable because physicians are well trained to support their patients satisfactorily and confidently in the increasingly complex digitalisation of healthcare. The ultimate goal of the educational community is the closest approach of students to the issues of telemedicine and e-health, the creation of a spirit of trust, and the acceptance and transmission of essential knowledge [268].

Noor also moved in this direction, seeking to discover the gaps in Saudi education for digital transformation in health [248]. The growing complexity of healthcare systems worldwide and the growing reliance of the medical profession on information technology for precise practices and treatments require specific standardised training in Information Technology (IT) health planning. Accreditation of core Information Technology (IT) is advancing internationally. Noor A. examined the state of Information Technology health programmes in the Kingdom of Saudi Arabia (KSA) to determine (1) how well international standards are met and (2) what further development is required in the light of recent initiatives of the Kingdom of Saudi Arabia on e-health [248]. Of the 109 institutions that participated in his research, only a few offered programmes specifically in Health Information Technology. As part of Saudi Vision 2030, Saudi digital transformation was deemed an urgent need. This initiative calls for applying internationally accepted Information Technology skills in education programmes and healthcare practices, which can only happen through greater collaboration between medical and technology educators and strategic partnerships with companies, medical centres and government agencies.

Another study by Diviani N. et al. adds to the knowledge of e-health education, demonstrating how online health information affects a person’s overall behaviour and enhances patients’ ability to understand, live and prepare for various health challenges. The increasing digitalisation of communication and healthcare requires further research into the digital divide and patients’ relationships with health professionals. Healthcare professionals must recognise the online information they seek and engage with patients to evaluate online health information and support joint healthcare-making [235].

## 6. Discussion

The selected studies comprise a conceptual model based on bibliographic research. Using an open-ended technique, we analyse the selected 287 articles, which are grouped into categories based on their context. This methodology provides readers with a good indication of issues concerning the timeliness of health digitalisation. A limitation of the methodology is that selected criteria of the method might be subjective in terms of the search terms and how the papers are selected. The articles indicate that this field is initial, and further research is needed. Although several articles have created a theoretical basis for corporate sustainability and strategic digital management, only limited studies provided guidelines on the strategic digital transformation process and its health implementation stages. However, studies have also developed sustainable models, software or applications in this area. This is also the reason for creating opportunities for future researchers, who will be closed to investigate this gap and improve the viability of digital health strategies. In addition, any work carried out in case studies provides fruitful results by facilitating researchers through deep penetration into sustainable digitalisation. No generalised frameworks are available to guide the wording and implementation of digital action plans. Thus, the need for quantitative or qualitative research is created, providing conclusions on the impact of internal or external factors in the sustainability process, implementation, adoption, planning, and challenges of digital health solutions in general, as well as the impact of digital transformation. Most existing studies explore the issue of digitalisation in a particular part of a nursing institution or a disease rather than the management strategy perspective. In this way, researchers ignore a debate on obstacles and problems that often face in practice during integration. Such an analysis could lead to more profound knowledge. 

## 7. Conclusions

In conclusion, our research observed a timeless analysis of systematised studies focusing on digital health developments. These studies broaden the researchers’ vision and provide vital information for further investigation. This article focuses on understanding digitalisation in healthcare, including, for the most part, the digitalisation of information and adopting appropriate parameters for further development. To build a more holistic view of digital health transformation, there is a great need for research on the management implications of digitalisation by different stakeholders. Finally, the development of telemedicine, the further enhancement of digital security and the strengthening of technological information systems will contribute to the universal acceptance of the digital health transformation by all involved.

## Figures and Tables

**Figure 1 ijerph-20-03407-f001:**
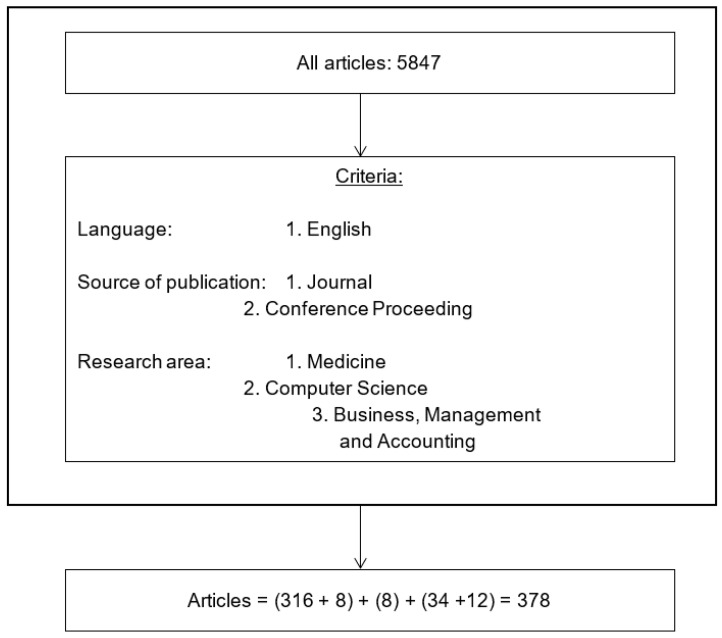
The diagram for the first phase of the selection process.

**Figure 2 ijerph-20-03407-f002:**
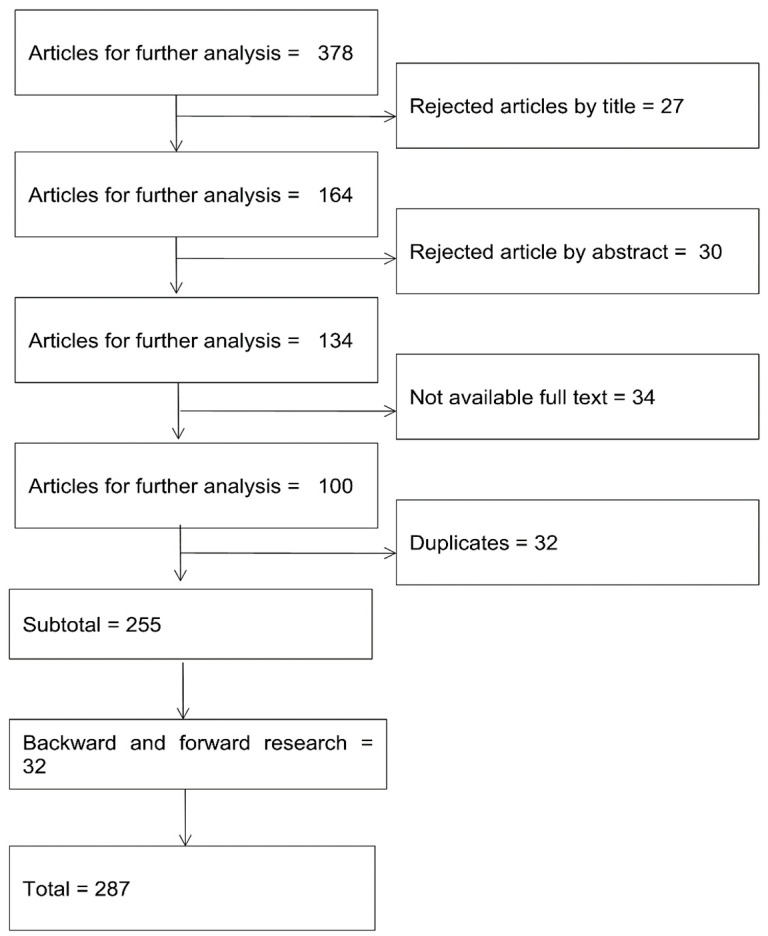
The diagram of the article selection process.

**Figure 3 ijerph-20-03407-f003:**
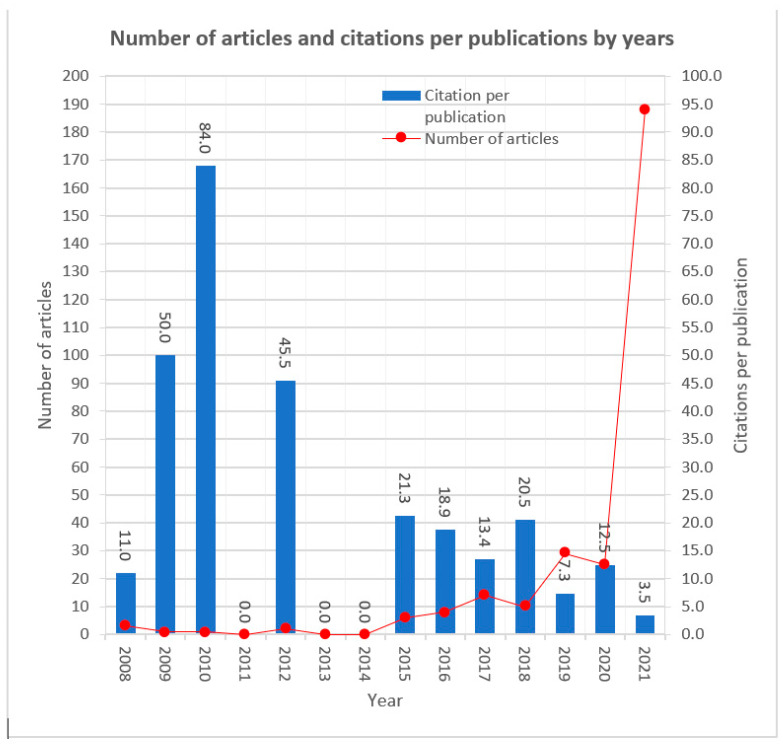
Number of articles and citations per publication by year.

**Figure 4 ijerph-20-03407-f004:**
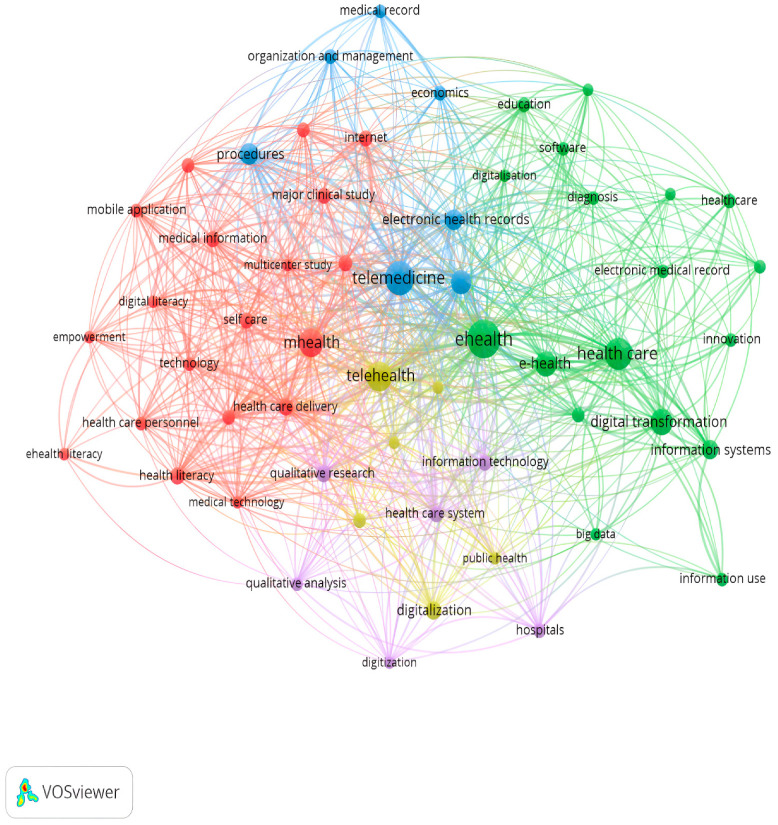
Bibliometric map of the digital transformation and healthcare.

**Figure 5 ijerph-20-03407-f005:**
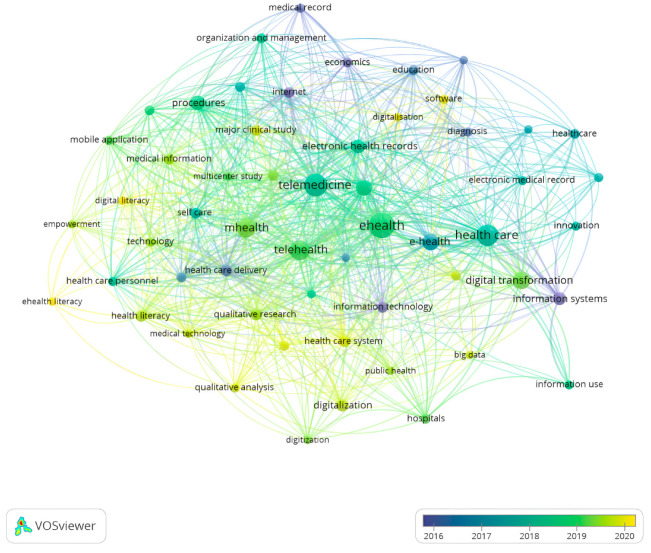
Network visualisation of keywords per year.

**Figure 6 ijerph-20-03407-f006:**
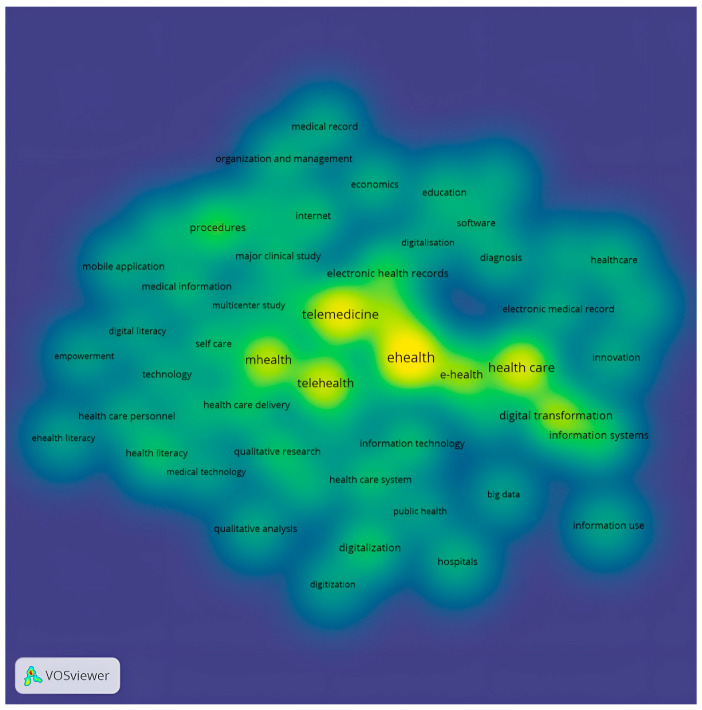
Heat map of keywords.

**Figure 7 ijerph-20-03407-f007:**
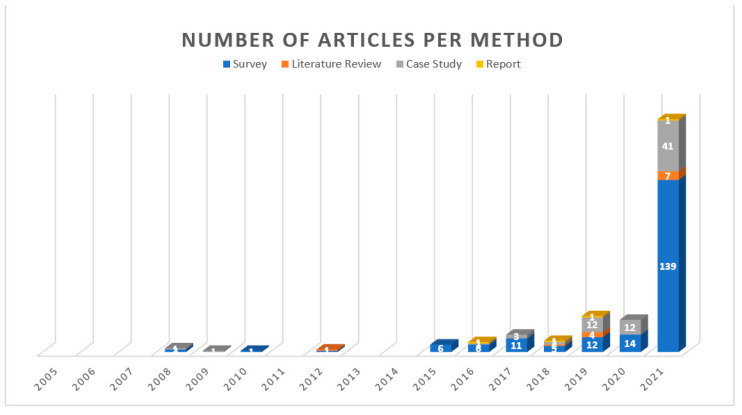
The map of number of articles per method for each year.

**Table 1 ijerph-20-03407-t001:** Previous Bibliographic Reviews.

	Reference	Keywords	Methodology	Results
1.	Kraus, S., et al., Digital transformation in healthcare: Analyzing the current state-of-research [9]	Digital* AND healthcare	2 Databases (EBSCO)—130 articles (ELSEVIER Science Direct and Springer Link)—340 articles	The article assesses how multiple stakeholders implement digital technologies for management and business purposes.
2.	Marques, Isabel C.P. and Ferreira, Joao J.M. Digital transformation in the area of health: a systematic review of 45 years of evolution. Health and Technology. 2020, 10, pp. 575–586. [8]	Digital AND Health AND Information System AND Management AND Hospital	1 Database (Scopus)—749 articles	Explore the potential of existing digital solutions to improve the quality and safety of healthcare and analyse the emerging trend of digital medicine.
3.	Kolasa, K. and G. Kozinski, How to Value Digital Health Interventions? A Systematic Literature Review [10]	Mhealth Mobile health Telemedicine Health app Wearables	3 Databases (Pubmed, Scopus and Science Direct)—34 articles	It proposed five recommendations for the generation of evidence to be considered in developing digital health solutions and suggestions for adopting the methodological approach in DHIs’ pricing and reimbursement.
4.	Mehdi Hosseinzadeh, Omed Hassan Ahmed, Ali Ehsani, Aram Mahmood Ahmed, Hawkar kamaran Hama. The impact of knowledge on e-health: a systematic literature review of the advanced systems [11]	Knowledge health Knowledge e-health	6 Databases (Google Scholar, Public Libraries, Science Direct, Springer Link, Web of Science and IEEE Xplore)—132 articles	Knowledge is considered one of the important research directions for many purposes in e-health.
5.	Shah Nazir, Yasir Ali, Naeem Ullah and Ivan Garcia—Magarino. Internet of Things for Healthcare Using Effects of Mobile Computing: A Systematic Literature Review, Hindawi, Wireless Communications and Mobile Computing, Volume 2019. [12]	(Internet of things OR IoT) AND (Smart hospitals) AND (Healthcare) AND (Mobile Computing) OR “Internet of things OR IoT” and “Smart hospitals” and “healthcare” and Mobile computing.”	5 Databases (Science Direct, Springer, IEEE, Taylor and Francis, Hindawi)—116 articles	Mobile computing extends the functionality of IoT in the healthcare environment by bringing massive support in the form of mobile health (m-health). In this research, a systematic literature review protocol is proposed to study how mobile computing assists IoT applications in healthcare, contributes to the current and future research of IoT in the healthcare system, brings privacy and security to health IoT devices, and affects the IoT in the healthcare system. Furthermore, the paper intends to study the impacts of mobile computing on IoT in the healthcare environment or intelligent hospitals.
6.	Chiranjeev Sanyal, Paul Stolee, Don Husereau. Economic evaluations of eHealth technologies: A systematic review, PLoS ONE [13]	Assistive technology Socially assistive robots Mobile health Mobile robot Smart home system Telecare Telehealth Telemedicine Wander prevention systems Mobile locator devices Gps Location-based technology Mobile apps Mobile application Cell phone Web-based Internet M-health M-health eHealth e-health older adult elderly seniors older patient cost-effective cost-utility economic evaluation	5 Databases (MEDLINE, EMBASE, CINAHL, NHS EED, and PsycINFO)—14 articles	E-health technologies can be used to provide resource-efficient patient-oriented care. This review identified the growing use of these technologies in managing chronic diseases in study populations, including older adults.
7.	Kampmeijer, R., et al., The use of e-health and m-health tools in health promotion and primary prevention among older adults: a systematic literature review. [14]	(“aged”[MeSH Terms] OR “aged”[All Fields] OR “elderly”[All Fields] OR “old”[All Fields] OR “senior”[All Fields] OR “seniors”[All Fields]) AND (“health promotion”[MeSH Terms] OR “health promotion”[All Fields] OR “promotion”[All Fields] OR “primary prevention”[MeSH Terms] OR “primary prevention”[All Fields] OR “prevention”[All Fields]) AND (“telemedicine”[MeSH Terms] OR “telemedicine”[All Fields] OR “telemedicine”[All Fields] OR “telehealth”[All Fields] OR “telehealth”[All Fields] OR “m-health”[All Fields] OR “m-health”[All Fields] OR “e-health”[All Fields] OR “e-health”[All Fields])	1 Database (PubMed)—45 articles	E-health and m-health tools are used by older adults in diverse health promotion programmes but also outside formal programmes to monitor and improve their health.
8.	Iyawa, G.E., M. Herselman, and A. Botha, Digital health innovation ecosystems: From a systematic literature review to conceptual framework [1]	Digital health Innovation Digital ecosystems	4 Databases (ACM, Science Direct, IEEE Xplore and SpringerLink)—65 articles	The study identified components of digital health, components creation relevant to the healthcare domain, and components of digital ecosystems.
9.	Gagnon, M.-P., et al., m-Health adoption by healthcare professionals: a systematic review. [15]	m-Health healthcare professionals and adoption	4 Databases (PubMed, Embase, Cinhal, PsychInfo)—33 articles	The Main perceived adoption factors to m-health at the individual, organisational, and contextual levels were the following: perceived usefulness and ease of use, design and technical concerns, cost, time, privacy and security issues, familiarity with the technology, risk-benefit assessment, and interaction with others (colleagues, patients, and management).
10.	Leslie W., Kim, A. and D. Szeto, The evidence for the economic value of ehealth in the united states today: a systematic review. J Int Soc Telemed EHealth, 2016. [16]	(telemedicine OR “Mobile Health” OR “Health, Mobile” OR mHealth OR mHealths OR Telehealth OR eHealth) AND (“Cost-Benefit Analysis” OR “Analyses, Cost-Benefit” OR “Analysis, Cost-Benefit” OR “Cost-Benefit Analyses” OR “Cost Benefit Analysis” OR “Analyses, Cost Benefit” OR “Analysis, Cost Benefit” OR “Cost Benefit Analyses” OR “Cost Effectiveness” OR “Effectiveness, Cost” OR “Cost-Benefit Data” OR “Cost Benefit Data” OR “Data, Cost-Benefit” OR “Cost-Utility Analysis” OR “Analyses, Cost-Utility” OR “Analysis, Cost-Utility” OR “Cost Utility Analysis” OR “Cost-Utility Analyses” OR “Economic Evaluation” OR “Economic Evaluations” OR “Evaluation, Economic” OR “Evaluations, Economic” OR “Marginal Analysis” OR “Analyses, Marginal” OR “Analysis, Marginal” OR “Marginal Analyses” OR “Cost Benefit” OR “Costs and Benefits” OR “Benefits and Costs” OR “CostEffectiveness Analysis” OR “Analysis, CostEffectiveness” OR “Cost-Effectiveness Analysis”) Virtual healthcare	2 Databases (PubMed and The Cochrane Library) -20 articles	The goal of this study is to evaluate the published economic evidence for e-health in the United States, analyse how well it supports the growth of the current e-health environment, and suggest what evidence is needed.
11.	Hu, Y. and G. Bai, A Systematic Literature Review of Cloud Computing in Ehealth. Health Informatics—[17]	(Cloud) AND (eHealth OR “electronic health” OR e-health)	5 Databases (ACM Digital Library, IEEE Xplore, Inspec, ISI Web of Science and Springer)—44 articles	With the unique superiority of the cloud in big data storage and processing ability, a hybrid cloud platform with mixed access control and security protection mechanisms will be the main research area for developing a citizen-centred home-based healthcare system.
12.	Boonstra, A., A. Versluis, and J.F.J. Vos, Implementing electronic health records in hospitals: a systematic literature review. BMC Health Services Research, 2014. 14(1): p. 370. [18]	“Electronic Health Record*” + implement* + hospital* “Electronic Health Record*” + implement* + “healthcare” “Electronic Health Record*” + implement* + clinic* “Electronic Patient Record*” + implment* + hospital* “Electronic Patient Record*” + implement* + “healthcare” “Electronic Patient Record*” + implement* + clinic* “Electronic Medical Record*” + implement* + hospital* “Electronic Medical Record*” + implement* + “healthcare” “Electronic Medical Record*” + implement* + clinic* “Computeri?ed Patient Record*” + implement* + hospital* “Computeri?ed Patient Record*” + implement* + “health care” “Computeri?ed Patient Record*” + implement* + clinic* “Electronic Health Care Record*” + implement* + hospital* “Electronic Health Care Record*” + implement* + “health care” “Electronic Health Care Record*” + implement* + clinic* “Computeri?ed Physician Order Entry” + implement* + hospital* “Computeri?ed Physician Order Entry” + implement* + “health care” “Computeri?ed Physician Order Entry” + implement* + clinic*	3 Databases (Web of Knowledge, EBSCO and the Cochrane Library)—21 articles	Although EHR systems are anticipated to affect hospitals’ performance positively, their implementation is complex.
13.	Pagliari, C., et al., What Is eHealth (4): A Scoping Exercise to Map the Field. J Med Internet Res, 2005. 7(1) [19]	“Ehealth OR e-health OR e*health”	8 Databases (Medline [PubMed], the Cumulative Index of Nursing and Allied Health Literature [CINAHL], the Science Citation Index [SCI], the Social Science Citation Index [SSCI], the Cochrane Library Database (including Dare, Central, NHS Economic Evaluation Database [NHS EED], Health Technology Assessment [HTA] database, NHS EED bibliographic) and Index to Scientific and Technical Proceedings (ISTP, now known as ISI Proceedings)—387 articles	Definitions of e-health vary concerning the functions, stakeholders, contexts, and theoretical issues targeted.

**Table 2 ijerph-20-03407-t002:** Search Strategy.

	Database	Search within	Keywords	No Sources
1.	Scopus	Article title, Abstract, Keywords	(Digital transformation or digitalization) AND (Ehealth or e-health or mhealth or m-health or healthcare) AND (health economics)	408
		Article title, Abstract, Keywords	(Digital transformation) AND (health)	1.152
2.	Science Direct	Article title	(Digital transformation) AND (health)	2.142
3.	PubMed	Article title, Abstract	(Digital transformation or digitalization) AND (Ehealth or e-health or mhealth or m-health or healthcare) AND (health economics)	978
		Article title	(Digital transformation) AND (health)	1.167
	Total			5.847

**Table 3 ijerph-20-03407-t003:** Concept Matrix Table.

No.	Author	Year	Method	Sample	Data Analysis	Concepts				
						Information Technology in Health	Education Impact of E-Health	Acceptance of E-Health	Telemedicine	Security of E-Health
1	Kesavadev, J, et al., [20]	2021	Case Study						Χ	
2	Attila, SZ et al., [21]	2021	Survey			Χ				
3	Malachynska, M et al., [22]	2021	Case Study			Χ				
4	Lu, WC et al., [23]	2021	Survey			Χ				
5	Burmann, A et al., [24]	2021	Case Study			Χ				
6	Bogumil-Ucan, S et al., [25]	2021	Case Study			Χ				
7	Zanutto, O [26]	2021	Survey			Χ				
8	Alauddin, MS; et al., [27]	2021	Survey			Χ				
9	Alterazi, HA [28]	2021	Survey			Χ				
10	Schmidt-Kaehler, S et al., [29]	2021	Case Study			Χ				
11	Zhao, Y et al., [30]	2021	Case Study			Χ	Χ			
12	Roth, CB et al., [31]	2021	Systematic Literature Review			Χ			Χ	
13	Ali, NA et al., [32]	2021	Case Study			Χ				
14	Alimbaev, A et al., [33]	2021	Case Study			Χ				
15	Dick, H et al., [34]	2021	Systematic Literature Review			Χ			Χ	
16	Alt, R et al., [35]	2021	Survey	a Vice President	-	Χ				
17	Bartosiewicz, A et al., [36]	2021	Survey			Χ			Χ	
18	Mussener, U [37]	2021	Survey					Χ		
19	Naumann, L et al., [38]	2021	Case Study	59 qualitative telephone interviews	The findings hinted at five priorities of e-health policy making: strategy, consensus-building, decision-making, implementation and evaluation that emerged from the stakeholders’ perception of the e-health policy.	Χ				
20	Saetra, HS et al., [39]	2021	Case Study			Χ				
21	Zoltan, V et al., [40]	2021	Survey			Χ				Χ
22	Hoch, P et al., [41]	2021	Survey						Χ	
23	De Vos, J [42]	2021	Survey			Χ				
24	Beaulieu, M et al., [43]	2021	Survey			Χ				
25	Dang, TH et al., [44]	2021	Survey			Χ	Χ		Χ	
26	Kraus, S et al., [9]	2021	Systematic Literature Review			Χ		Χ	Χ	
27	Gauthier, P et al., [45]	2021	Survey					Χ		
28	Zhang, JS et al., [46]	2021	Survey			Χ				
29	Mallmann, CA et al., [47]	2021	Survey	513 breast cancer patients from 2012 to 2020	Statistical analysis	Χ				
30	Fons, AQ [48]	2021	Survey			Χ				
31	Chatterjee, S et al., [49]	2021	Survey	Consumers of different age groups & people working in the healthcare sector (including doctors)	Qualitative analysis	Χ	Χ			
32	Wasmann, JWA et al., [50]	2021	Survey			Χ				
33	Kanungo, RP et al., [51]	2021	Survey			Χ				
34	Fernandez-Luque, L et al., [52]	2021	Survey			Χ				
35	Wilson, A et al., [53]	2021	Survey			Χ				
36	Ziadlou, D [54]	2021	Survey	US health care leaders	Qualitative analysis	Χ	Χ			
37	Oh, SS et al., [55]	2021	Survey					Χ	Χ	
38	Knitza, J et al., [56]	2021	Survey			Χ				
39	Sergi, D et al., [57]	2021	Survey			Χ				
40	Rosalia, RA et al., [58]	2021	Case Study			Χ				
41	[Anonymous] [59]	2021	Survey			Χ				
42	Prisyazhnaya, NV et al., [60]	2021	Survey				Χ			
43	Odone, A et al. [61]	2021	Case Study	Variety of participants	Qualitative and quantitative analysis	Χ				
44	Balta, M et al., [62]	2021	Case Study			Χ		Χ		
45	Mues, S et al., [63]	2021	Survey						Χ	
46	Frick, NRJ et al., [64]	2021	Case Study	Physicians (nine female and seven male experts)	Thematic analysis	Χ				
47	Dendere, R et al., [65]	2021	Survey			Χ				
48	Neumann, M et al., [66]	2021	Survey	The dean or the most senior academic individual responsible for the medical curriculum development	Descriptive statistics in Microsoft Excel (Version 16.38)	Χ				
49	Su, Y et al., [67]	2021	Case Study			Χ				
50	Masuda, Y et al., [68]	2021	Survey			Χ				
51	Frennert, S [69]	2021	Survey			Χ	Χ			
52	Hasselgren, A et al., [70]	2021	Survey			Χ				Χ
53	Kim, HK et al., [71]	2021	Survey			Χ		Χ		
54	Marchant, G et al., [72]	2021	Survey	569 adults	Statistical analysis	Χ				
55	Malfatti, G et al., [73]	2021	Survey			Χ				
56	Krasuska, M et al., [74]	2021	Case Study	628 interviews, observed 190 meetings and analysed 499 documents	Thematical analysis	Χ				
57	Piccialli, F et al., [75]	2021	Survey			Χ				
58	Kyllingstad, N et al., [76]	2021	Survey			Χ				
59	Frasquilho, D et al., [77]	2021	Case Study			Χ				
60	Leone, D et al., [78]	2021	Case Study			Χ				
61	Kwon, IWG et al., [79]	2021	Report				Χ			
62	Sim, SS et al., [80]	2021	Systematic Literature Review						Χ	
63	Christie, HL et al., [81]	2021	Case Study	Experts (n = 483) in the fields of e-health, dementia, and caregiving were contacted via email	Qualitative analysis	Χ				
64	Eberle, C et al., [82]	2021	Survey	2887 patients	Qualitative analysis	Χ				
65	Popkova, EG et al., [83]	2021	Survey			Χ				
66	Reich, C et al., [84]	2021	Survey			Χ				
67	Hanrieder, T et al., [85]	2021	Survey			Χ				
68	Aleksashina, AA et al., [86]	2021	Survey			Χ			Χ	
69	Haase, CB et al., [87]	2021	Survey			Χ				
70	Mishra, A et al., [88]	2021	Survey			Χ				
71	Kokshagina, O [89]	2021	Survey			Χ				
72	Loch, T et al., [90]	2021	Survey						Χ	
73	Cajander, A et al., [91]	2021	Survey	17 interviews with nurses (*n* = 9) and physicians (*n* = 8)	Thematical analysis	Χ			Χ	
74	Botrugno, C [92]	2021	Survey				Χ			
75	Jacquemard, T et al., [93]	2021	Survey			Χ				
76	Behnke, M et al., [94]	2021	Survey			Χ				
77	Peltoniemi, T et al., [95]	2021	Case Study							Χ
78	Glock, H et al., [96]	2021	Survey					Χ		
79	Weitzel, EC et al., [97]	2021	Survey					Χ		
80	Sullivan, C et al., [98]	2021	Case Study			Χ				
81	Luca, MM et al., [99]	2021	Survey					Χ		
82	Negro-Calduch, E et al., [100]	2021	Systematic Literature Review			Χ				
83	Werutsky, G et al.,Denninghoff, V et al., [101]	2021	Survey			Χ				
84	Piasecki, J et al., [102]	2021	Survey			Χ	Χ			
85	Broenneke, JB et al., [103]	2021	Survey			Χ				
86	Faure, S et al., [104]	2021	Survey						Χ	
87	Ghaleb, EAA et al., [105]	2021	Survey			Χ		Χ		
88	Verket, M et al., [106]	2021	Survey			Χ				
89	Lenz, S [107]	2021	Survey	15 interviews with persons from different areas of digital health care	Theoretical sampling	Χ				
90	De Sutter, E et al., [108]	2021	Survey	31 healthcare professionals active	Qualitative analysis	Χ				
91	Gevko, V et al., [109]	2021	Survey			Χ				
92	El Majdoubi, D et al., [110]	2021	Survey			Χ				
93	Thakur, A et al., [111]	2021	Case Study			Χ				
94	Persson, J et al., [112]	2021	Survey			Χ				
95	Zippel-Schultz, B et al., [113]	2021	Survey	49 patients and 33 of their informal caregivers.	Qualitative analysis				Χ	
96	Lam, K et al., [114]	2021	Survey			Χ				
97	Manzeschke, A [115]	2021	Survey			Χ				
98	Dyda, A et al., [116]	2021	Case Study			Χ			Χ	
99	Beckmann, M et al., [117]	2021	Case Study	Variety of participants	Qualitative and quantitative analysis					Χ
100	Numair, T et al., [118]	2021	Survey	Kenya: Interviewees included nurses, community health workers, and operators hired exclusively for data entry in the WIRE system. Laos: As no operators were hired in Lao PDR, interviewees included nurses, doctors, and midwives who used the WIRE system daily. (20 healthcare workers in Kenya & Laos PDR)	Qualitative and quantitative analysis	Χ				
101	Xiroudaki, S et al., [119]	2021	Case Study			Χ				
102	Droste, W et al., [120]	2021	Survey						Χ	
103	Lee, JY et al., [121]	2021	Systematic Literature Review			Χ				
104	Giovagnoli, et al., [122]	2021	Survey			Χ				
105	Daguenet, et al., [123]	2021	Survey						Χ	
106	Hubmann, et al., [124]	2021	Survey			Χ				
107	Vikhrov, et al., [125]	2021	Survey			Χ				
108	Jahn, HK et al., [126]	2021	Survey	198 complete and 45 incomplete survey responses from physicians	Statistical analysis	Χ				
109	Low et al., [127]	2021	Survey			Χ				
110	Levasluoto, et al., [128]	2021	Case Study	23 interviews	Thematical analysis	Χ				
111	Verma, et al., [129]	2021	Survey			Χ				
112	Leung, PPL et al., [130]	2021	Case Study			Χ				
113	Weber, S et al., [131]	2021	Survey			Χ				
114	Hogervorst, S et al., [132]	2021	Survey	Patients (11), group HCPs (5 + 6), interviews HCPs (4)	Thematical analysis	Χ				
115	Khan, ich et al., [133]	2021	Systematic Literature Review			Χ				
116	Cherif, et al., [134]	2021	Survey					Χ		
117	Bingham, et al., [135]	2021	Survey	19 registered nurses	Descriptive statistics	Χ				
118	Broich, et al., [136]	2021	Survey			Χ				
119	Klemme, et al., [137]	2021	Survey	The study consisted of 15 semi-structured interviews with academic staff (*n* = 7 professors and postdoctoral researchers, three female, four male) in the field of intelligent systems and technology in healthcare and staff at practice partners (*n* = 8 heads of department, two female, six male) in healthcare technology and economy (a hospital, a digital innovation and engineering company and a manufacturer of household appliances) and social institutions (foundations and aid organisations for people with disabilities).	Qualitative analysis	Χ	Χ			
120	Dillenseger, et al., [138]	2021	Survey			Χ				
121	Wangler, et al., [139]	2021	Survey			Χ				
122	Kuhn, et al., [140]	2021	Survey	Students (35)	Qualitative analysis		Χ			
123	Aldekhyyel, et al., [141]	2021	Survey						Χ	
124	Christlein, et al., [142]	2021	Survey			Χ				
125	Bergier, et al., [143]	2021	Survey						Χ	
126	Sitges-Macia, et al., [144]	2021	Survey			Χ				
127	Rani, et al., [145]	2021	Survey				Χ			
128	Fredriksen, et al., [146]	2021	Case Study	Healthcare employees from a volunteer centre and from municipality healthcare units in three municipalities	Qualitative analysis	Χ				
129	Caixeta, et al., [147]	2021	Survey			Χ				
130	Gupta, et al., [148]	2021	Survey			Χ				
131	Dobson, et al., [149]	2021	Survey			Χ				
132	Choi, K et al., [150]	2021	Survey						Χ	
133	Muller-Wirtz, et al., [151]	2021	Case Study			Χ				
134	Sembekov, et al., [152]	2021	Survey			Χ				
135	Aulenkamp, et al., [153]	2021	Survey			Χ	Χ			
136	Paul, et al., [154]	2021	Survey	16 key stakeholders	Thematical analysis	Χ				
137	Lemmen, et al., [155]	2021	Survey	62 citizens and 13 patients	Qualitative analysis	Χ				
138	Golz, et al., [156]	2021	Survey			Χ				
139	Tarikere, et al., [157]	2021	Survey			Χ				
140	Li, et al., [158]	2021	Case Study			Χ				
141	Rouge-Bugat, et al., [159]	2021	Case Study			Χ				
142	Iodice, et al., [160]	2021	Survey						Χ	
143	Kulzer, B [161]	2021	Survey			Χ				
144	Khosla, et al., [162]	2021	Survey			Χ				
145	Dantas, et al., [163]	2021	Survey			Χ				
146	Gaur, et al., [164]	2021	Survey			Χ				
147	Khodadad-Saryazdi, A [165]	2021	Case Study				Χ	Χ	Χ	
148	Bellavista, et al., [166]	2021	Case Study			Χ				
149	Laukka, et al., [167]	2021	Case Study					Χ	Χ	
150	Singh, et al., [168]	2021	Survey			Χ				
151	Patalano, et al., [169]	2021	Survey			Χ				
152	Mantel-Teeuwisse, et al., [170]	2021	Survey			Χ				
153	Mues, et al., [171]	2021	Survey						Χ	
154	Bosch-Capblanch, et al., [172]	2021	Survey							Χ
155	Jaboyedoff, et al., [173]	2021	Survey	336 common data elements (CDEs)	Qualitative analysis	Χ				
156	Nadhamuni, et al., [174]	2021	Survey			Χ				
157	Hertling, et al., [175]	2021	Survey						Χ	
158	Khan, et al., [176]	2021	Survey			Χ				
159	Mun, et al., [177]	2021	Survey			Χ			Χ	
160	Xi, et al., [178]	2021	Survey				Χ			
161	Weichert, et al., M [179]	2021	Survey			Χ				
162	Liang, et al., [180]	2021	Survey				Χ			
163	Williams, et al., [181]	2021	Survey	508 interviews, 163 observed meetings, and analysis of 325 documents.	Qualitative analysis—Sociotechnical principles, combining deductive and inductive methods		Χ			
164	Feroz, et al., [182]	2021	Case Study			Χ				
165	Huser, et al., [183]	2021	Case Study			Χ				
166	Apostolos, K [184]	2021	Survey			Χ				
167	Simsek, et al., [185]	2021	Survey			Χ			Χ	
168	Khamisy-Farah, et al., [186]	2021	Survey			Χ				
169	Egarter, et al., [187]	2021	Case Study				Χ			
170	Can, et al., [188]	2021	Survey			Χ				
171	Sung, et al., [189]	2021	Survey	278 e-logbook database entries and 379 procedures in the hospital records from 14 users were analysed. Interviews with 12 e-logbook users found overall satisfaction.	Statistical analysis			Χ		Χ
172	Zoellner, et al., [190]	2021	Survey						Χ	
173	Oliveira, et al., [191]	2021	Case Study	Recipients numbering 151 (21% of the universe) completed the questionnaire: trade (49), industry (41), services (28), health (15), and education (18).	Quantitative analysis	Χ				
174	Goudarzi, et al., [192]	2021	Survey							Χ
175	Li, et al., [193]	2021	Survey						Χ	Χ
176	Klimanov, et al., [194]	2021	Case Study			Χ				
177	Nadav, et al., [195]	2021	Survey	Eight focus group interviews were conducted with 30 health and social care professionals	Qualitative analysis			Χ		
178	Spanakis, et al., [196]	2021	Survey			Χ				
179	Polyakov, et al., [197]	2021	Survey					Χ		
180	Fristedt, et al., [198]	2021	Survey	Intervention group (*n* = 80) & control group (*n* = 80)	Data will be coded and manually entered in SPSS	Χ				
181	Mandal, et al., [199]	2021	Survey			Χ				
182	Ozdemir, V [200]	2021	Survey			Χ				
183	Eberle, et al., [201]	2021	Survey						Χ	
184	Iakovleva, et al., [202]	2021	Case Study			Χ				
185	von Solodkoff, et al., [203]	2021	Survey	In the questionnaire, the participants (*n* = 217). A total of 27 subjects (mean age 51 years, min: 23 years, max: 86 years) participated in the interviews.	Statistical analysis				Χ	
186	Khuntia, et al., [204]	2021	Survey			Χ			Χ	
187	Ochoa, et al., [205]	2021	Survey			Χ				
188	Masłoń-Oracz, et al., [206]	2021	Case Study			X		X		
189	Abrahams, et al., [207]	2020	Survey			X	X			
190	Agnihothri, et al., [208]	2020	Survey			X				
191	Bukowski, et al., [209]	2020	Survey			X				X
192	Chiang, et al., [210]	2020	Survey					X		X
193	Cobelli, et al., [211]	2020	Survey	Pharmacists (82)	Qualitative content analysis	X				
194	Crawford, et al., [212]	2020	Survey			X		X		
195	Gjellebæk, et al., [2]	2020	Case Study	Employees and middle managers	Thematic analysis	X				
196	Nascimento, et al., [213]	2020	Case Study			X				
197	Geiger, et al., [214]	2020	Case Study	Specialist in neurosurery & resident (296)	Statistical Analysis	X			X	
198	Eden, et al., [4]	2020	Survey	Medical, nursing, allied health, administrative and executive roles (92)	Analysis of Cohen’s kappa (k)	X		X		
199	Gochhait, et al., [215]	2020	Case Study			X			X	
200	Kernebeck, et al., [216]	2020	Case Study			X				
201	Klinker, et al., [217]	2020	Survey	Staff of health care facilities (14)	Microsoft HoloLens, Vuzix m100			X		
202	Krasuska, et al., M.; Williams, R.; Sheikh, A.; Franklin, B. D.; Heeney, C.; Lane, W.; Mozaffar, H.; Mason, K.; Eason, et al., [218]	2020	Survey	Staff of health care facilities (113)	Qualitative analysis	X				
203	Leigh, et al., [219]	2020	Survey			X				
204	Minssen, et al., [220]	2020	Survey			X				
205	Mueller, et al., [221]	2020	Case Study	Staff of health care facilities (20)	Qualitative analysis	X			X	
206	Nadarzynski, et al., [222]	2020	Case Study	Patients (257)	Statistical analysis	X			X	
207	Pekkarinen, et al., [223]	2020	Case Study	Variety of participants (24)	The analytical framework is based on Nardi and O’Day’s five components of information ecology: system, diversity, co-evolution, keystone species, and locality.	X				
208	Rajamäki, et al., [224]	2020	Survey							X
209	Salamah, et al., [225]	2020	Case Study			X				
210	Stephanie, et al., [226]	2020	Survey			X				
211	Sultana, et al., [227]	2020	Survey			X				X
212	Visconti, et al., [228]	2020	Case Study			X				
213	Yousaf et al., [229]	2020	Case Study					X		
214	Asthana, et al., [230]	2019	Survey			X				
215	Astruc, B. [231]	2019	Case Study			X			X	
216	Baltaxe, et al., [232]	2019	Report			X				
217	Caumanns, J. [233]	2019	Case Study			X				
218	Diamantopoulos, et al., [234]	2019	Case Study					X		X
219	Diviani, et al., [235]	2019	Survey	Variety of participants (165)	Qualitative analysis		X			
220	EYGM [236]	2019	Survey					X		
221	Hatzivasilis, et al., [237]	2019	Survey							X
222	Go Jefferies, et al., [238]	2019	Case Study			X			X	
223	Kivimaa, P., et al., [239]	2019	Systematic Literature Review				X			
224	Klocek, A., et al., [240]	2019	Case Study	Variety of people (153)	Statistical analysis	X				
225	Kohl, S., et al., [241]	2019	Survey							X
226	Kouroubali, et al., [242]	2019	Case Study			X		X		
227	Manard, et al., [243]	2019	Case Study			X				
228	Mende M. [244]	2019	Survey			X				
229	Mishra et al., [245]	2019	Systematic Literature Review			X	X	X		
230	Niemelä, et al., [246]	2019	Survey	Health professionals, child patients’ parents, and the healthcare industry	Systematically analysed according to the process structure (pre-, intra-, post-surgery, and home care).	X				
231	Nittas, V., et al. [247]	2019	Survey							X
232	Noor, A. [248]	2019	Case Study	Students and Staff in colleges and universities	Qualitative analysis		X			
233	Pape, L., et al. [249]	2019	Case Study					X		
234	Patrício, et al., [250]	2019	Survey			X				
235	Russo Spena, T., Cristina, M. [251]	2019	Survey			X				
236	Rydenfält, C., et al., [252]	2019	Case Study	Variety of people (264)	NVivo 10 (QSR International, Melbourne, Australia)			X		
237	Savikko, et al., [253]	2019	Case Study			X				
238	Vial, G [254]	2019	Systematic Literature Review				X			
239	Wangdahl, J.M., et al., [255]	2019	Case Study	Variety of people (600)	Binary logistic regression analysis		X			
240	Watson, et al., [256]	2019	Systematic Literature Review			X				
241	Weigand, et al., [257]	2019	Survey							X
242	Zanutto, A. [258]	2019	Survey	Staff of health care facilities (6836)	Qualitative analysis			X		
243	Eden, et al., [3]	2018	Systematic Literature Review			X				
244	Goh, W., et al. [259]	2018	Survey							X
245	Kayser, L., et al., [260]	2018	Survey				X			
246	Poss-Doering, R. et al., [261]	2018	Case Study	Patients (11) & Doctors (3)	Statistical analysis	X		X		X
247	Khatoon, et al., [262]	2018	Survey					X		X
248	Melchiorre, M.G., et al., [263]	2018	Case Study			X				
249	Ngwenyama, et al., [264]	2018	Survey			X				
250	Öberg, U.A.-O., et al., [265]	2018	Survey	Primary healthcare nurses (20)	Qualitative analysis			X		
251	Parkin, et al., [266]	2018	Report				X			
252	Tuzii, J., [267]	2018	Case Study			X				
253	Brockes, C., et al., [268]	2017	Survey	Students (28)	Mann–Whitney U-Test		X		X	
254	Cavusoglu, et al., [269]	2017	Survey					X		
255	Cerdan, et al., [270]	2017	Case Study	Patients (29)	Qualitative analysis			X		
256	Coppolino, et al., [271]	2017	Survey							X
257	Geiger, et al., [272]	2017	Survey			X				
258	Giacosa, et al., [273]	2017	Survey				X			
259	Hong, et al., [274]	2017	Survey			X				
260	Hüsers, J., et al., [275]	2017	Case Study	Nurses (534)	All data were analysed using R (Version 3.2.1)	X				
261	Parviainen, et al., [276]	2017	Survey			X				
262	Paulin, A. [277]	2017	Survey							X
263	Schobel, J., et al. [278]	2017	Survey							X
264	Seddon, et al., [279]	2017	Survey			X				
265	Thorseng, et al., [280]	2017	Survey	Variety of participants	Qualitative analysis	X				
266	Tuzii, J. [267]	2017	Case Study			X				
267	Amato, F., et al., [281]	2016	Survey							X
268	Bongaerts, et al., [282]	2016	Survey					X		
269	Cucciniello, et al., [283]	2016	Survey			X				
270	Evans, R.S. [284]	2016	Survey			X				
271	Faried, et al., [285]	2016	Report					X		
272	Harjumaa, M., et al., [286]	2016	Survey	Various organisations (12)	Interview data was then analysed thematically.					X
273	Mattsson, T., [287]	2016	Case Study							X
274	Mazor, et al., [288]	2016	Survey			X				
275	Anwar, et al., [289]	2015	Survey					X		X
276	Kostkova, P., [290]	2015	Survey			X				
277	Laur, A., [291]	2015	Survey							X
278	Sultan, N., [292]	2015	Survey			X	X			
279	Nudurupati, et al., [293]	2015	Survey			X				
280	Sanders, K., et al., [294]	2015	Survey	Healthcare professionals (17)	Qualitative analysis	X				
281	Cook, et al., [295]	2012	A Systematic Literature Review				X			
282	Khan, et al., [296]	2012	Survey					X		
283	Agarwal, R., et al., [5]	2010	Survey			X				
284	Thomas, et al., [297]	2009	Case Study							X
285	Buccoliero, et al., [298]	2008	Survey			X				
286	Hikmet, et al., [299]	2008	Case Study	Variety of participants	Quantitive analysis	X				
287	Zdravković, S. [300]	2008	Survey			Χ			X	

## Data Availability

Not applicable.

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
