# Peer review of "Digital Transformation in Healthcare: Technology Acceptance and Its Applications"

_ijerph, 2023, doi:10.3390/ijerph20043407_

Round 1
Reviewer 1 Report
Dear authors, I read the article entitled Digital Transformation in Healthcare: Technology acceptance 2 and its applications with great interest. The topic is a current and relevant topic. Readers of IJERPH might profit from your article. However, I have some suggestions for the revision of the manuscript and I encourage you to re-submit your article.
Abstract:
In the abstract, you outline that you are going to “analyse the changes taking place in the field of 11 healthcare due to digital transformation”. This should be specified by adding aspects you are investigating. In addition, please clarify why digital transformation has touched every aspect of human.
Overall, please be a little more specific about the information you provide in the abstract.
Introduction:
In the first line of the introduction you use the expression “The digital transformation of health”. Please reconsider using this expression as no overall digital transformation of health can be stated.
Further, the term transformation is used a lot, however, no conceptual discussion of transformation takes place. What does transformation mean? What is the difference to innovation? How is transformation to be classified? Please add a brief theoretical discussion of the concept of (digital) transformation from the perspective of transformation research or the sociology of technology.
Results and discussion:
Please reconsider using 3D diagrams or too much colors. Further, some of the illustrations are hard to read especially referring to figure 6 to 8. In addition, please reconsider moving some of the illustrations to the appendix. Please provide in the main text only the relevant information. Table 3 and 4 are too large for the main text. Consider a condensed form.
Conclusions:
Compared to the very good results, the conclusions are far too brief. Please consider to include recommendations for educational providers in health care.
Author Response
Comment 1
Dear authors, I read the article entitled Digital Transformation in Healthcare: Technology acceptance 2 and its applications with great interest. The topic is a current and relevant topic. Readers
of IJERPH might profit from your article. However, I have some suggestions for the revision of the manuscript and I encourage you to re-submit your article.
Answer 1.
Thank you for taking the time to read our paper, and you’re encouraging us to re-submit our article. Thank you for your good words for our first draft
*************************************************************************************************************************
Comment 2
Abstract:
In the abstract, you outline that you are going to “analyse the changes taking place in the field of 11 healthcare due to digital transformation”. This should be specified by adding aspects you are investigating. In addition, please clarify why digital transformation has touched every aspect of humans.
Overall, please be a little more specific about the information you provide in the abstract.
Answer 2.
Thank you for your comment, which helped us to improve the abstract. We rewrite the abstract entirely. We clearly stated the motivation for this paper. We precisely explained the methodology we follow to produce results. We also mention the significance of our work and the importance of the results.
************************************************************************************************************************
Comment 3
Introduction:
In the first line of the introduction you use the expression “The digital transformation of health”. Please reconsider using this expression as no overall digital transformation of health can be stated.
Further, the term transformation is used a lot, however, noconceptual discussion of transformation takes place. What does transformation mean? What is the difference to innovation? How is transformation to be classified? Please add a brief theoretical discussion of the concept of (digital) transformation from the perspective of transformation research or the sociology of technology.
Answer 3
Thank you for your comment, which helped us to improve the introduction. We changed the introduction to emphasise the importance of digital transformation and the positive consequences for healthcare systems. In this way, we introduce the readers to the importance of our work. On the other hand, our purpose is to perform the literature review for digital transformation in healthcare
************************************************************************************************************************
Comment 4
Results and discussion:
Please reconsider using 3D diagrams or too much colors. Further, some of the illustrations are hard to read especially referring to figure 6 to 8. In addition, please reconsider moving some of the illustrations to the appendix. Please provide in the main text only the relevant information. Table 3 and 4 are too large for the main text. Consider a condensed form.
Answer 4
Thank you for your comment, which helped us to improve the results and discussion. We replaced the problematic figures with new figures with higher resolutions (figures 6 to 8).
Table 3 and Table 4 are the main results of the methodology by Wester and Watson (MIS Quarterly, 26(2): xiii-xxiii, 2002). Table 3 consists of 287 articles which were derived for the selection process. Table 4 is the concept matrix table which is the result of the methodology and includes the articles’ categories.
********************************************************************************************************************
Comment 5
Conclusions:
Compared to the very good results, the conclusions are far too brief. Please consider to include recommendations for educational providers in health care.
Answer 5
Thank you for your comment. The conclusion should be brief. Education depends on the need of the organization. We have discussed the educational impact of e-health in section 5.5.
***************************************************************************************************************************

Reviewer 2 Report
The purpose of this article is to provide an assessment of the current literature on digital health transformation, as well as to identify potential vulnerabilities that make its implementation impossible. The article is an interesting one but the authors need to attend to some comments to make it more relevant and suitable for the targeted audience:
· The authors should let us know what led to the motivation of this research
· Authors mention clinicians often report side effects using digital technologies and health professionals oppose using digital systems/technologies. However, they don´t explain some of them. This is an interesting topic that is not explained to the audience.
· Format of bibliographic references are different. For example, G. E. et al. (2016) (line 38) and [1] (line 40).
· There are some mistakes regarding grammar, typos and English language.
· In general, the quality of the figures is poor and all of them should be changed.
· Figures 6, 7 and 8: the keywords are blurred, they are unreadable.
· Paragraph 1 of subsection "3.1 Categories of articles" describes the categorization of the articles. However, the referred Figure (Figure 3) shows the chronological development of the publications (very similar to Figure 4). It would be interesting a Figure that represents the chronological development of the publications by categories instead.
· It’s not clear how the network analysis was created (what are the inputs? What is the aim of the articles categorization mentioned in subsection “3.1 Categories of artibles”? How the keywords were created/selected? Why was there a limit to the number of individual words?)
· Figures 8 and 6 are redundant
· Figure 9 is not referred in the text and the number of the articles (in some years) are unreadable. Figures 5 and 9 shows similar information, for this reason, Figure 9 does not provide much information
· In Section 3.2, authors explain that they add 32 articles but this fact is not described in Figure 2 which explain how they collected the articles. This information should be added to Figure 2 and the authors should be explained this in section 2.2.2 Initial Search
· Tables 3, 4 and 5 are excessively long and very difficult to read. It is recommended to make a summary and summarize the articles for a better understanding.
· It lacks the sufficient discussions and conclusions
· The limitations of this study are not discussed
Author Response
Thank you

Reviewer 3 Report
The paper is an interisting overview and analysis of exixting literature in the issue of digital transformation in health.
As major observation the authors should explain why thay have not included Google Scholar and Web of Science (by Clarivate).
The authors should better explain how they arrive from more than 5000 articles to 378 as first screening; then they should be coherent in the whole paper: in discussion they wrote 278, probably a digit error for 378 or 255?
The description of the method of network analysis should be in Methods, and if appropriate explain if the distances between words or intensities of lines express the intensity of relations and how.
The resolution of the networf figures must be enhanced.
REsults for some concepts are actually too short, and table 4 and 5 could be as supplementary. Those tables increase the load of the paper and results are followed very bad, with interruptions and recall. The Result chapter could be more descriptive and fluent.
The chapter related to Security need more deep discussion. Security of data, privacy of patients are important issues, and something should be discussed on regulation about those issues.
The last major is that in my opinion the authors have stated four questions in introduction, but thay haven't give the answer to those, but they have opened new questions. I agree with questions that they put, but I think that an attempt to give answere to the four of the introduction should be done. Successively it's useful to open new problem.
Minor problems:
at line 57, a little explanation of side effects of technology;
at line 434, the concept should be rewritten, because the repetition of words doesn't help in getting your comment.
Author Response
Comment 0
The paper is an interesting overview and analysis of existing literature in the issue of digital transformation in health.
Answer
Thank you for taking the time to read our paper, and you’re encouraging us to re-submit our article. Thank you for your good words for our first draft
Comment 1
As major observation the authors should explain why they have not included Google Scholar and Web of Science (by Clarivate).
Answer
All the papers included in the web of science are also in Scopus. Very rarely we have a document which belongs to web of science rather than Scopus. Google scholar can include prestigious journals like Scopus
Comment 2
The authors should better explain how they arrive from more than 5000 articles to 378 as first screening; then they should be coherent in the whole paper: in discussion, they wrote 278,probably a digit error for 378 or 255?
Answer
This is not a mistake. We explain in section the following:
The search was performed on the following databases: Scopus, Science Direct, and PubMed, using the keywords "digital transformation," "digitalisation," "Ehealth or e-health," "mhealth or m-health," "healthcare," and "health economics". We selected publications from the search of international journals and conference proceedings. We collected papers from 2008 until 2021. The documents sought belonged to strategy, management, computer science, medicine, and health professions. Finally, the published works were in English only. The total number of articles collected using the keywords as shown in Table 2 was 5,847.
Look the justification for Figure 1 and figure 2 and their explanations.
Comment 3
The description of the method of network analysis should be in Methods, and if appropriate explain if the distances between words or intensities of lines express the intensity of relations and how.
Answers
We added section 2.2 in methodology which explains the philosophy of network graphs
Comment 4a
The resolution of the networf figures must be enhanced.
Answer
Thank you for your comment, which helped us to improve the results and discussion. We replaced the problematic figures with new figures with higher resolutions (figures 6 to 8).
Comment 4b
REsults for some concepts are actually too short, and table 4 and 5 could be as supplementary. Those tables increase the load of the paper and results are followed very bad, with interruptions and recall. The Result chapter could be more descriptive and fluent.
Answer
Table 4 and 5 are main results Thank you for your comment, which helped us to improve the results and discussion. We replaced the problematic figures with new figures with higher resolutions (figures 6 to 8).
Table 3 and Table 4 are the main results of the methodology by Wester and Watson (MIS Quarterly, 26(2): xiii-xxiii, 2002). Table 3 consists of 287 articles which were derived for the selection process. Table 4 is the concept matrix table which is the result of the methodology and includes the articles’ categories.
Comment
The chapter related to Security need more deep discussion. Security of data, privacy of patients are important issues, and something should be discussed on regulation about those issues.
Answer
Thank you for this comment. We decided not to develop this further.
Comment
The last major is that in my opinion the authors have stated four questions in introduction, but thay haven't give the answer to those, but they have opened new questions. I agree with questions that they put, but I think that an attempt to give answere to the four of the introduction should be done . Successively it's useful to open new problem.
Answer
WE have removed these questions which is not the main aim for this paper.
Round 2
Reviewer 1 Report
Dear authors, thank you for revising the manuscript. I have no further comments.
Author Response
Thank you.